# The Prognostic Significance of Epidermal Growth Factor Receptor Amplification and Epidermal Growth Factor Receptor Variant III Mutation in Glioblastoma: A Systematic Review and Meta-Analysis with Implications for Targeted Therapy

**DOI:** 10.3390/ijms26083539

**Published:** 2025-04-09

**Authors:** Fangge Zhu, Jinming Qiu, Haoyuan Ye, Wenting Su, Renxi Wang, Yi Fu

**Affiliations:** Laboratory of Brain Disorders, Beijing Institute of Brain Disorders, Ministry of Science and Technology, Collaborative Innovation Center for Brain Disorders, Capital Medical University, No. 10 Xitoutiao, Beijing 100069, China; zfg1019@mail.ccmu.edu.cn (F.Z.); xqxxps@mail.ccmu.edu.cn (J.Q.); yhy1019@mail.ccmu.edu.cn (H.Y.); wtsu@ccmu.edu.cn (W.S.)

**Keywords:** glioblastoma (GBM), EGFR amplification, EGFRvIII mutation, prognostic biomarker

## Abstract

Glioblastoma (GBM) is the most aggressive and heterogeneous neoplasm among central nervous system tumors, with a dismal prognosis and a high recurrence rate. Among the various genetic alterations found in GBM, the amplification of epidermal growth factor receptor (EGFR) and the EGFR variant III (EGFRvIII) mutation are among the most common, though their prognostic value remains controversial. This systematic review and meta-analysis aimed to provide a comprehensive evaluation of the diagnostic and prognostic significance of EGFR amplification and the EGFRvIII mutation in GBM patients, incorporating recent studies published in the past few years to offer a more complete and up-to-date analysis. An extensive search of the PubMed, Web of Science, and Scopus databases was conducted, including studies that reported on EGFR and/or the EGFRvIII mutation status with detailed survival data. A total of 32 studies with 4208 GBM patients were included. The results indicated that EGFR amplification significantly correlated with worse OS (Overall survival) (HR = 1.27, 95% CI: 1.03–1.57), suggesting that EGFR amplification is an independent prognostic marker. The prognostic value of EGFRvIII was inconclusive, with a pooled hazard ratio for overall survival of 1.13 (95% CI: 0.94–1.36), indicating no significant effect on survival in the general population. However, a subgroup analysis suggested that EGFRvIII may be associated with poorer outcomes, particularly in recurrent GBM patients, where its prognostic significance became more evident. Furthermore, subgroup analyses based on geographic region revealed significant heterogeneity in the prognostic impact of EGFR amplification across different populations. In American cohorts, EGFR amplification was strongly associated with an increased risk of mortality (HR = 1.53, 95% CI: 1.28–1.84, *p* = 0.001), suggesting that it serves as a more reliable prognostic marker in this region. In contrast, no significant prognostic impact of EGFR amplification was observed in Asian (HR = 0.64, 95% CI: 0.35–1.17) or European (HR = 0.98, 95% CI: 0.80–1.19) populations. Overall, this study underscores the potential of EGFR amplification as a prognostic marker in GBM, while further research is needed to fully elucidate the role of the EGFRvIII mutation, particularly in specific patient subgroups. Clarifying these associations could offer important insights for targeted treatment strategies, improving patient outcomes.

## 1. Introduction

Glioblastoma (GBM) is the most common and aggressive malignant brain tumor, characterized by poor survival rates, high recurrence, and extensive heterogeneity. It accounts for over 50% of all brain tumors diagnosed annually. The prognosis for GBM patients remains dismal, with a median overall survival of approximately 12–18 months, a 5-year survival rate of 5.1%, and a 1-year survival rate of 37.2% [1]. The current basic therapy for treating GBM is surgical resection whenever possible, followed by a combination therapy of radiotherapy and temozolomide (TMZ) chemotherapy, which is termed the “Stupp protocol”. A pivotal study by Stupp demonstrated that the combination of TMZ and radiotherapy extended survival by 2.5 months compared to monotherapy [2].

A frequent genetic alteration in GBM is the amplification of the epidermal growth factor receptor (EGFR), a transmembrane tyrosine kinase receptor. Nearly 50% of GBMs exhibit EGFR overexpression, and 50–60% of these cases also harbor the EGFRvIII mutant [3]. EGFR is a member of the ErbB family, and as a proto-oncogene, it plays a critical role in promoting survival, proliferation, invasion, and angiogenesis in GBM, while also enhancing the resistance to radiotherapy and chemotherapy [4]. The EGFRvIII mutation, the most common variant of EGFR, has been shown to further increase the invasiveness and proliferation of GBM cells [5,6]. In a GBM model derived from neural stem cells with engineered Nf1/Pten co-deletion and EGFRvIII overexpression, aggressive tumor growth and infiltration were observed compared to an Nf1/Pten-only tumor [7]. Many studies have demonstrated that the EGFRvIII mutant typically occurs alongside EGFR amplification, suggesting that these two molecules cooperate to activate downstream signaling pathways. In comparison to cells expressing either EGFR or EGFRvIII alone, the co-expression of both EGFR and EGFRvIII significantly promoted GBM xenograft growth in the LN-229 nude mouse model [8]. A more recent study has highlighted the cooperative role of EGFR and EGFRvIII in driving GBM progression through the activation of the ROCK2/WNT/TLR2 signaling axis [9].

Several molecules are overexpressed in GBM compared to healthy tissue, making them potential biomarkers for predicting the clinical prognosis. These biomarkers have been identified and validated in numerous studies using both mouse models and clinical trials. However, due to the complexity and inconsistency across studies, particularly in clinical trials, the definitive link between these biomarkers and the prognosis remains controversial. EGFR amplification, the most common alteration in GBM, has long been considered an oncogenic driver of glioma genesis, as demonstrated in many studies. Yet, its prognostic and diagnostic relevance remains disputed. For instance, a multivariate analysis of 185 newly diagnosed GBM patients at the Washington University School of Medicine in St. Louis found that EGFR amplification had an independent prognostic value [10]. Conversely, a recent prospective cohort study of GBM patients failed to demonstrate a prognostic link with the EGFR status [11]. Similarly, the prognostic value of the EGFRvIII mutation in GBM remains unclear, with limited or incomplete clinical data yielding inconsistent results. Therefore, a systematic review and meta-analysis based on the most up-to-date data is crucial for clarifying the prognostic significance of EGFR amplification and the EGFRvIII mutant in GBM.

In this study, we performed a comprehensive search of the PubMed, Web of Science, and Scopus databases to identify and analyze relevant clinical trials to evaluate whether EGFR amplification or the EGFRvIII mutation could serve as independent prognostic predictors for GBM. Our goal was to evaluate the potential of EGFR amplification and the EGFRvIII mutation as independent prognostic biomarkers for GBM. By systematically reviewing studies published over the past two decades, we aimed to provide an updated and comprehensive understanding of their diagnostic and prognostic significance. Additionally, we sought to explore potential factors that could influence the clinical utility of these markers, such as treatment regimens and geographic variations. This approach is intended to offer deeper insights into the prognostic implications of EGFR-related alterations, with the aim of refining targeted therapeutic strategies for GBM patients.

## 2. Materials and Methods

### 2.1. Search Strategy

This systematic review and meta-analysis was designed and carried out according to the Preferred Reporting Items for Systematic Reviews and Meta Analyses (PRISMA) guidelines [12] (Appendix A). This study was registered in advance on the PROSPERO website under the registration number CRD42025634135. The PubMed, Web of Science, and Scopus databases were searched for eligible articles published until November 2024. The search strategy was systematically expansive and based on various combinations of the following keywords: “Glioblastoma OR GBM”, “EGFRvIII OR epidermal growth factor receptor variant III”, “EGFR OR epidermal growth factor receptor”. The search strategy employed both subject terms and unrestricted keywords in different permutations. The detailed search strategies are available in the Appendix A.

### 2.2. Inclusion and Exclusion Criteria

The inclusion criteria for this meta-analysis were as follows: (1) articles that reported on patients with GBM who were reported to have an EGFR and/or EGFRvIII mutation status; (2) articles in which detailed survival data were reported, including the overall survival (OS), progression-free survival (PFS), hazard ratios (HRs), and other mutant states.

The studies deemed ineligible for inclusion in the meta-analysis were (1) articles with incomplete survival data; (2) articles that reported on patients diagnosed with pediatric glioblastoma multiforme (pGBM); (3) duplicate articles; (4) cell and animal experiments; and (5) case reports, letters, meeting abstracts, comments, editorials, and personal correspondence.

### 2.3. Data Extraction

Two investigators independently extracted data from all the eligible publications, and discrepancies were resolved through discussion. Standardized forms were built and utilized for data extraction, recording information such as the first author, publication year, country, study design, patient population, and survival outcomes (OS and PFS). The OS refers to the duration of time from the start of a study (such as the diagnosis or the initiation of treatment) to death from any cause. PFS is defined as the time from the initiation of a study to the occurrence of disease progression (e.g., tumor growth or recurrence) or death from any cause. An Engauge Digitizer (Version 12.1) was utilized to extract the data from Kaplan–Meier curves when OS or PFS rates were not explicitly reported in the paper, and the HR value were estimated using the methods described by Tierney et al. [13].

### 2.4. Quality Assessment and Risk of Bias Evaluation

The Newcastle–Ottawa scale (NOS) was used to independently assess the quality of the included studies [14]. The quality of the randomized controlled trials (RCTs) was assessed using the Cochrane Collaboration Risk of Bias Tool 2 (ROB 2). A quality assessment was conducted independently by two reviewers, and a third reviewer was consulted for any uncertainties. The risk of a publication bias was assessed through a funnel plot analysis including >10 studies, generated using Review Manager (Version 5.3). Additionally, Egger’s test and Begg’s test were performed using Stata (Version 18.0) to further evaluate the risk of a publication bias among the included studies.

### 2.5. Statistical Analysis

The study outcomes included OS and PFS. To investigate the effect of EGFRvIII/EGFR mutations on patient survival, a meta-analysis was conducted by pooling the HRs for OS and PFS, along with their 95% confidence intervals (CIs). To investigate the sources of heterogeneity, we performed a subgroup analysis, stratifying the studies by factors such as geographic regions and patient characteristics. Statistical significance was defined as a two-sided *p*-value of <0.05 or a pooled HR with a 95% CI that excluded 1. Conventionally, an HR of greater than 1 indicates that mutations are associated with worse patient prognoses. The heterogeneity was assessed using the I^2^ statistic, with thresholds of 0–25%, 25–50%, 50–75%, and 75–100% indicating no, moderate, large, and extreme heterogeneity, respectively. A random-effects model was used when I^2^ exceeded 50%, while a fixed-effects model was employed for supplementary analysis to evaluate robustness. When the number of included studies is small, a fixed-effects model is employed for the analysis. Sensitivity analysis using the leave-one-out approach assessed the influence of individual studies on the pooled estimates. The Z-test determined the statistical significance of pooled HRs, with results considered significant at a *p*-value < 0.05. All analyses were performed using Review Manager 5.3 and Stata 18.0.

## 3. Results

### 3.1. Literature Screening

Figure 1 shows the study selection procedure. By searching through three different databases, a total of 151 articles were reviewed. After eliminating duplicates, 127 articles were included. A total of 83 articles were excluded after an initial screening of titles and abstracts due to their apparent irrelevance. Subsequently, 44 articles were selected for a full-text review, of which 27 met the predefined inclusion criteria. Furthermore, 5 additional studies identified through manual searches were included, yielding a final dataset of 32 studies [10,11,15,16,17,18,19,20,21,22,23,24,25,26,27,28,29,30,31,32,33,34,35,36,37,38,39,40,41,42,43,44].

### 3.2. Data Extraction and Quality Assessment

An overview of the patient characteristics and relevant survival metrics from the included studies was presented in Table 1. The included studies were published between 2004 and 2024 and exclusively involved glioblastoma patients. Of the 32 included studies, 30 were cohort studies, and 2 were randomized controlled trials [28,38]. Among them, 20 studies provided survival data for patients with EGFRvIII mutations [16,19,20,21,22,23,26,28,29,31,32,33,34,35,37,38,39,42,43,44], while 14 studies reported survival data for patients with EGFR amplification [10,11,15,17,18,20,21,22,24,27,36,40,41,42]. Additionally, nine studies investigated newly diagnosed glioblastoma [23,28,29,32,35,38,42,43,44], and four focused on recurrent cases [19,22,31,34]. Radiotherapy combined with temozolomide (TMZ) was the most common treatment strategy, with some studies investigating monoclonal antibody therapies for recurrent glioblastoma. Considering geographic population differences, four studies focused on Asian populations [11,26,34,44], ten on American populations [10,15,16,18,19,22,23,24,35,41], and sixteen on European populations [17,19,20,21,27,29,31,32,33,36,37,38,40,42,43,44].

The HR for the OS associated with the EGFRvIII mutation ranged from 0.442 to 3.695 (95% CI: 0.209–0.714 and 0.651–20.963, respectively). Similarly, the HR for the OS associated with EGFR amplification ranged from 0.64 to 3.92 (95% CI: 0.35–1.16 and 1.03–14.9, respectively). The study quality was assessed using the NOS scale for the 30 cohort studies, with scores of six or higher indicating high quality (Table 2). For the two randomized controlled trials, the Cochrane ROB 2 tool (version 2019) was applied to evaluate the risk of bias (Table 3 and Table 4). Overall, the included studies demonstrated an acceptable methodological quality.

### 3.3. Meta Analysis

#### 3.3.1. No Significant Association Between EGFRvIII Status and OS in GBM

This meta-analysis integrated the survival data on the association between the EGFRvIII status and the overall survival from 20 studies [16,19,20,21,22,23,26,28,29,31,32,33,34,35,37,38,39,42,43,44], comprising a total of 2571 patients, including 1030 EGFRvIII-positive and 1541 EGFRvIII-negative cases. The analysis demonstrated no significant association between EGFRvIII positivity and overall survival (pooled HR = 1.13, 95% CI: 0.94–1.36, *p* = 0.20) (Figure 2a). The heterogeneity analysis indicated substantial heterogeneity among the included studies (Tau^2^ = 0.09, Chi^2^ = 46.63, I^2^ = 59%, *p* = 0.0004), which may be attributed to variations in the timing and spatial expression of EGFRvIII, treatment regimens, follow-up durations, or other study-specific factors. To ensure the robustness of these findings, a fixed-effects model analysis was conducted, yielding results consistent with those obtained using the random-effects model (Appendix A). Although the pooled HR (1.13) is greater than one, the 95% confidence interval (0.94–1.36) includes one, indicating no statistically significant impact of EGFRvIII positivity on overall survival, based on the current evidence. This suggests that the current evidence is limited by heterogeneity and insufficient statistical power. Future research with standardized, high-quality designs is crucial to clarify this association and its clinical implications.

#### 3.3.2. EGFR Amplification Correlates with Worse Prognosis in GBM

A total of 14 studies were included to evaluate the impact of EGFR amplification on survival outcomes in glioblastoma patients [10,11,15,17,18,20,21,22,24,27,36,40,41,42], involving 580 patients with EGFR amplification (EGFR-amp) and 1032 patients with wildtype EGFR (EGFR-wt). Among these studies, eight directly reported hazard ratios for overall survival, comparing EGFR-amp and EGFR-wt [10,11,20,22,27,36,41,42], while survival data for the remaining six studies were extracted from Kaplan–Meier survival curves [15,17,18,21,24,40]. The heterogeneity analysis revealed moderate heterogeneity (Tau^2^ = 0.07, Chi^2^ = 26.75, I^2^ = 51%, *p* = 0.02). Given the observed heterogeneity, a random-effects model was applied to estimate the combined effect size. The analysis revealed a significant association between EGFR amplification and reduced overall survival in the GBM patients (pooled HR = 1.27, 95% CI 1.03–1.57, *p* = 0.02) (Figure 2b). A supplementary fixed-effects model analysis yielded consistent findings (Appendix A). Overall, the findings indicate a significant association between EGFR amplification and reduced prognoses in GBM patients, suggesting its potential role as a prognostic biomarker.

#### 3.3.3. EGFRvIII Status or EGFR Amplification Show No Significant Association with PFS in GBM

Six studies were included to analyze the pooled HR for PFS between EGFRvIII-positive and -negative glioblastoma patients [19,26,28,31,38,43]. The heterogeneity analysis indicated moderate heterogeneity among the included studies (I^2^ = 57%, *p* = 0.04) (Figure 3a). No statistically significant association was observed between EGFRvIII status and progression-free survival in the GBM patients, as shown by the random-effects model (HR = 1.06, 95% CI: 0.70–1.61) and the fixed-effects model (HR = 0.90, 95% CI: 0.73–1.11) (Appendix A). Collectively, the current evidence does not establish a clear association between the EGFRvIII status and PFS in glioblastoma patients.

Five studies were included to analyze the association between EGFR amplification and PFS in glioblastoma patients [10,11,25,27,30]. The heterogeneity analysis demonstrated substantial variability among the studies (I^2^ = 65%, *p* = 0.02). The pooled analysis showed no statistically significant association between EGFR amplification and PFS, as indicated by both the random-effects model (HR = 1.23, 95% CI: 0.77–1.96) (Figure 3b) and the fixed-effects model (HR = 1.17, 95% CI: 0.92–1.48) (Appendix A). These findings may be affected by the small number of included studies and the observed heterogeneity across them.

#### 3.3.4. No Significant Impact of Co-Occurring EGFR Amplification and EGFRvIII Mutation on OS in GBM

To analyze the association between the co-occurrence of EGFR amplification and the EGFRvIII mutation and overall survival in GBM patients, five studies were selected from the 32 included articles [16,28,38,39,42]. In these studies, all the patients evaluated for EGFRvIII-related survival were confirmed to have EGFR amplification. The analysis yielded a pooled HR of 0.94 (95% CI: 0.78–1.13, *p* = 0.49), indicating that the co-occurrence of EGFR amplification and the EGFRvIII mutation is not significantly associated with poorer overall survival in GBM patients (Figure 4a).

Additionally, six studies specifically investigated IDH-wildtype (IDH-wt) glioblastoma patients [19,20,22,29,34,39]. An exploratory meta-analysis of these studies demonstrated that the EGFRvIII mutation in IDH-wt glioblastoma patients is not significantly associated with poorer overall survival (HR = 1.02, *p* = 0.93) (Figure 4b).

#### 3.3.5. EGFRvIII Mutant and EGFR Amplification Are Associated with Poorer Survival in the Americas, but Not in Asia or Europe

To explore the source of heterogeneity, subgroup analyses were conducted based on populations to examine the overall survival differences across regions. Some studies primarily focused on Asian populations [26,34,44], while the study by Aldape et al. [16,20,22,23,35] focused on American populations, and the study by Chiesa et al. [19,21,29,31,32,33,37,38,42,43] focused on European populations. The subgroup analysis demonstrated that EGFRvIII-positive glioblastoma patients had a significantly higher risk of death in the American population (HR = 1.51, 95% CI: 1.02–2.24, *p* = 0.04). However, no significant associations were observed in the European population (HR = 1.12, 95% CI: 0.83–1.51, *p* = 0.46) or the Asian population (HR = 0.92, 95% CI: 0.64–1.32, *p* = 0.66) (Figure 5a). The random-effects model showed no significant subgroup differences (Chi^2^ = 3.25, *p* = 0.20, I^2^ = 38.5%), indicating that the impact of the EGFRvIII mutation on OS in GBM patients is unlikely to vary substantially by geographic region. However, the fixed-effects model (*p* = 0.02) suggested potential differences (Appendix A), highlighting the need for further investigation through larger, region-specific studies to validate these observations.

The subgroup difference analysis (*p* = 0.001) indicated statistically significant heterogeneity in the prognostic impact of EGFR amplification across different geographic regions, suggesting that the effect of EGFR amplification on overall survival varies significantly among populations. In the Americas, EGFR amplification was associated with a markedly increased risk of mortality (HR = 1.53, 95% CI: 1.28–1.84) (Figure 5b and Appendix A), indicating that EGFR amplification can serve as a stronger prognostic marker for GBM patients in this region. In contrast, no statistically significant impact of EGFR amplification on OS was observed in Asian [11] (HR = 0.64, 95% CI: 0.35–1.16) and European (HR = 0.98, 95% CI: 0.80–1.19) populations, suggesting limited prognostic value in these regions. These findings highlight that the prognostic impact of EGFR amplification on OS in GBM patients varies by geographic region, potentially reflecting region-specific genetic, environmental, or treatment-related factors.

#### 3.3.6. EGFRvIII Mutation Associated with Poorer Survival in Recurrent GBM

We also performed subgroup analyses based on the patient characteristics. Nine studies by Yang et al. [23,28,29,32,35,38,42,43,44] focused on newly diagnosed glioblastoma, while four studies by Nozawa et al. [19,22,31,34] targeted recurrent glioblastoma. Given the low heterogeneity among the studies on recurrent glioblastoma (I^2^ = 0%), a fixed-effects model was used for the subgroup analysis (Figure 6). The subgroup results revealed a significant difference in the impact of the EGFRvIII mutation on overall survival between newly diagnosed and recurrent glioblastoma patients (*p* = 0.005). The analysis demonstrated that the EGFRvIII mutation was significantly associated with an increased risk of death in recurrent glioblastoma patients (HR = 1.63, 95% CI: 1.15–2.31, *p* = 0.006), while no significant association was observed in newly diagnosed glioblastoma patients (HR = 0.96, 95% CI: 0.83–1.10, *p* = 0.56). To address potential heterogeneity, a random-effects model was also applied, confirming consistency with the fixed-effects model’s results (Appendix A). These findings suggest that the EGFRvIII mutation is a significant prognostic factor for poorer survival in recurrent glioblastoma but not in newly diagnosed cases.

#### 3.3.7. Sensitivity Analysis

Sensitivity analyses were conducted by sequentially removing each study to assess the robustness of the results (Appendix A). No single study had a significant influence on the overall findings, indicating that our analysis results are stable and reliable.

#### 3.3.8. Publication Bias

The funnel plot generated by RevMan 5.3 indicated that the distribution of the studies was symmetrical (Appendix A). Additionally, as the assessment of symmetry is subjective, we further evaluated the publication bias using Stata 18.0 software. The results of both Egger’s test and Begg’s test revealed no significant publication bias (Egger’s test: *p* = 0.096/Begg’s test: *p* = 0.347 for EGFRvIII; Egger’s test: *p* = 0.134/Begg’s test: *p* = 0.300 for EGFR). The high *p*-values from both tests suggest that the distribution of the effect sizes was unlikely to be influenced by a publication bias (Appendix A).

## 4. Discussion

Prognostic studies of molecular alterations in GBM provide crucial criteria for classifying the tumor grade, assessing the prognosis, and supporting the development of targeted therapies. To date, a variety of molecular biomarkers have been identified in GBM, including O^6^-methylguanine DNA methyltransferase (MGMT) gene promoter methylation, isocitrate dehydrogenase 1/2 (IDH1/2) mutations, telomerase reverse transcriptase (TERT) gene promoter mutations, and EGFR amplification or mutation. These biomarkers have demonstrated potential in predicting survival outcomes for GBM patients [45,46,47,48]. Among these potential biomarkers, EGFR amplification and the EGFRvIII mutant are the most common genetic alterations in GBM.

EGFR, a member of the receptor tyrosine kinase (RTK) family, consists of an extracellular, ligand-binding domain and an intracellular tyrosine kinase domain. Nearly 50% of GBM patients exhibit EGFR amplification, and 25–30% have the EGFRvIII mutant [49,50]. Several mechanisms have been elucidated to explain the roles of EGFR and its mutants in GBM proliferation, survival, and metabolism. EGFR activates the RAS signaling pathway by recruiting the GRB2/RAS/GEF/SOS complex, leading to the activation of the ERK1/2 signaling cascade, which modulates a range of proteins and transcription factors that are critical for tumorigenesis [45,51,52]. Moreover, STAT3 can be activated by multiple pathways, either directly or indirectly, through EGFR-mediated phosphorylation, further promoting GBM tumorigenesis [53,54]. Unlike wildtype EGFR, EGFRvIII lacks an EGF-binding domain but exhibits constitutively active kinase activity. The expression of EGFRvIII promotes the proliferation, angiogenesis, and invasion of GBM cells via traditional EGFR signaling pathways. More than 2000 enhancers have been found to be activated by EGFRvIII. Recent studies indicate that EGFRvIII often functions in conjunction with EGFR, forming dimers that enhance tumor growth [8,11]. This suggests that considering EGFR and EGFRvIII together may be more effective for understanding their combined roles and exploring therapeutic targets. While EGFR is a potent oncogene and its alterations are widely recognized as drivers of tumorigenesis in GBM, the correlation between EGFR alterations and patient survival remains a topic of debate. Recent retrospective and prospective studies on immune checkpoint inhibitors (ICIs) for GBM at Mount Sinai Hospital found that EGFR amplification was associated with worse survival outcomes for GBM patients treated with ICIs [22]. A multivariate analysis from the Washington University School of Medicine also indicated that EGFR mutations were independently associated with poor survival in GBM patients [10]. However, previous studies have shown inconsistent results.

While EGFR amplification has been linked to worse outcomes in some independent clinical cohorts, systematic meta-analyses have reported no significant association with overall survival. For instance, Heena Sareen’s 2022 systematic review and meta-analysis found no significant link between EGFR amplification or high expression and OS in GBM patients [55]. This analysis, based on five clinical studies involving 575 patients, concluded negatively, primarily due to incomplete data, compared to our study. Similarly, a 2015 meta-analysis also failed to establish a significant prognostic value for EGFR amplification in GBM, having included only three studies [56]. These limitations underscore the need for more robust clinical data to improve the accuracy of conclusions. In our systematic meta-analysis, which included 14 studies with a larger cohort of 1751 patients, we evaluated the prognostic significance of EGFR amplification in GBM. Interestingly, contrary to the findings of the 2015 and 2023 meta-analyses, our analysis results suggest a significant prognostic role for EGFR amplification in GBM. This finding provides an important reference for future studies investigating EGFR amplification and therapies targeted against GBM.

Additionally, we conducted a comprehensive update on the prognostic value of EGFRvIII mutations in GBM, including 20 studies with 2571 patients. Although the hazard ratio for EGFRvIII was 1.13, indicating a potential detrimental impact on OS, the 95% confidence interval of 0.94–1.36 suggested no statistically significant effect. Given the high heterogeneity among the included studies, our conclusion that the EGFRvIII mutation was not significantly associated with the GBM prognosis has limited clinical relevance. However, through a survival analysis in recurrent GBM patients, we found that the EGFRvIII mutation serves as a significant marker of a poor prognosis, specifically in this subgroup. This finding highlights the potential of EGFRvIII as a diagnostic tool for identifying high-risk recurrent GBM cases, which may benefit from more personalized and molecularly targeted therapies, such as EGFRvIII-directed CAR-T therapy [57,58]. Nonetheless, it is important to consider other possible influencing factors, such as treatment protocols, diagnostic timing, and coexisting molecular alterations, which could affect the observed prognostic value. Furthermore, variations in treatment regimens and mutation detection methods may contribute to the heterogeneity observed across studies. Although multiple animal studies have demonstrated that the EGFRvIII mutation enhances the malignant characteristics of GBM, increasing the tumor’s aggressiveness and resulting in ill-defined tumor margins, recent meta-analyses have not confirmed a significant association between the EGFRvIII mutation and poor prognoses in GBM patients. Our meta-analysis aligned with these studies. Moreover, the EGFRvIII mutation frequently coexists with EGFR amplification [59], and the coexistence of these two malignant alterations may indicate tumor progression and increased treatment challenges. Interestingly, our analysis suggests that the co-occurrence of EGFR amplification and the EGFRvIII mutation did not necessarily lead to worse overall survival in GBM patients. This contrasts with prior studies indicating a combined adverse effect of these alterations on prognoses [9], suggesting a need for further investigation into how EGFR amplification and EGFRvIII mutations interacted with other molecular and clinical factors, such as the tumor microenvironment and therapy resistance.

Global epidemiological data indicate that GBM incidence is highest in North America and Europe, with the non-Hispanic white population in the United States exhibiting the highest GBM prevalence [59,60,61], suggesting potential racial differences in GBM susceptibility. Our subgroup analysis further revealed significant geographical variations in the prognostic implications of EGFR alterations and EGFRvIII amplification in GBM. We propose that multiple factors contribute to these regional differences, including racial genetic variations, molecular epidemiology, treatment patterns, and environmental influences. These factors may account for the observed regional differences in the EGFR and EGFRvIII mutation rates observed among Caucasian populations. Future research should focus on conducting region-specific meta-analyses with more granular data on patient demographics, treatment protocols, and tumor molecular characteristics to validate these findings. Additionally, integrating multi-omics approaches, such as transcriptomics and proteomics, could provide deeper insights into the biological mechanisms underlying these regional differences. Recognizing these complexities will be crucial for optimizing personalized treatment strategies for GBM patients globally.

Currently, the standard-of-care treatment for primary GBM consists of surgical resection followed by radiation therapy, with concomitant and maintenance temozolomide. However, for recurrent GBM, no established standard therapy exists [62]. TMZ, a lipophilic agent capable of penetrating the blood–brain barrier, remains the gold standard for treating primary GBM [63]. In the clinical trials reviewed in this study, TMZ was used as an adjunctive therapy for almost all primary GBM patients. While the diagnostic utility of EGFR or EGFRvIII as biomarkers for GBM remains contentious, various therapies targeting EGFR and EGFRvIII have been explored, including tyrosine kinase inhibitors (TKIs), monoclonal antibodies, peptide vaccines, and chimeric antigen receptor T-cells [64]. Three generations of EGFR TKIs have been developed for GBM treatment [53]. The third-generation TKI, Osimertinib, specifically targets EGFR-activating mutations and the T790M-resistance mutation [53]. Our published study also demonstrated that Osimertinib effectively inhibits EGFR expression and the activation of associated pathways [65]. Monoclonal antibodies, such as Nimotuzumab and Cetuximab, both targeting the L2 domain of EGFR, have been tested in Phase I and II clinical trials. In a Phase II trial, 35 patients with anaplastic astrocytoma or GBM treated with Nimotuzumab combined with radiation therapy showed prolonged survival compared to the patients receiving radiation alone [66]. A study involving 43 recurrent GBM patients treated with Cetuximab in combination with bevacizumab and irinotecan demonstrated partial responses in 9 patients and a median overall survival of 29 weeks [67]. A bispecific antibody, bscEGFRvIIIxCD3, targeting EGFRvIII, has been shown to extend survival in U87-bearing mice while activating antitumor T-cell responses, with 75% of the mice achieving complete responses [68]. Additionally, recent reports on third-generation CAR-T therapy targeting HER2 (also known as EGFR2) have demonstrated effective antitumor activity and strong cytotoxicity against GBM cells [69]. A CAR-T targeting EGFRvIII was also developed, showing specific cytotoxic activity against SB28-EGFRvIII cells, presenting a novel strategy for GBM therapy [70].

In the latest WHO classification of CNS tumors, glioblastoma is defined as an IDH-wildtype grade IV diffuse glioma, which can be further subdivided into molecular subtypes based on specific characteristics [71]. According to extensive transcriptome profiling and high-throughput, next-generation sequencing, GBM is commonly classified into four subtypes with distinct molecular features, clinical behaviors, and prognoses: classical (CL), mesenchymal (MES), proneural (PN), and neural (which is often excluded due to the belief that it represents normal neuronal lineage contamination). MES tumors are characterized by high NF1 expression, while PN tumors frequently exhibit PDGFRA mutations and contain tumor cells similar to oligodendrocyte progenitors and neural progenitors. CL tumors mainly consist of astrocyte-like cells and are frequently associated with EGFR and EGFRvIII mutations [72,73]. CL-type GBM patients tend to respond relatively well to chemoradiotherapy but generally have poor prognoses [74]. EGFR alterations play a pivotal role in driving the differentiation of GBM tumors toward the CL subtype. As early as 1998, Sibilia et al. demonstrated that EGFR could promote astrocytes’ proliferation and differentiation in the brain [75]. In 2016, Lu et al. suggested that the negative regulation between Oligo2 and EGFR may facilitate the transformation of astrocytes [76]. In a retrospective study of 80 CL-type recurrent GBM patients, Hovinga et al. found that EGFR mutations were associated with poorer outcomes following bevacizumab treatment [25]. However, another study by Park et al. involving 23 recurrent GBM patients reported no significant correlation between EGFR mutations and OS (HR = 0.84, *p* = 0.585) [77]. Our analysis indicates that the EGFRvIII mutation is associated with a poorer prognosis in recurrent glioblastoma. Based on the association between EGFR alterations and the CL subtype, our findings are most likely relevant to the prognostic evaluation of the EGFRvIII mutation in CL-subtype recurrent GBM patients. However, its prognostic significance for other GBM subtypes remains uncertain. Among the studies we included, only one speculated that these patients might belong to the MES subtype, based on MAPK pathway activation in the NGS results [19]. The other studies lacked detailed GBM subtype information. The further analysis of the association between highly mutated oncogenes and the prognosis of different GBM subtypes remains necessary. Future studies should consider the potential intratumoral heterogeneity of molecular subtypes, investigate subtype-specific oncogenic alterations, and identify substitute markers. These factors are crucial for accurately predicting prognosis and optimizing treatment strategies for GBM patients with distinct molecular subtypes.

This study has certain limitations that need to be considered. Our included studies span from 2004 to 2024, during which time, the WHO definition of GBM underwent two significant changes. The first was in the 2016 WHO classification, where GBM was divided into the IDH-wildtype and IDH-mutant subtypes. The second was in the 2021 WHO classification, where IDH-mutant tumors were reclassified as anaplastic astrocytoma (WHO grade IV) instead of GBM. Among our included studies, three reported a small proportion of IDH-mutant cases (<10%) in GBM patients with EGFR amplification or the EGFRvIII mutation: Lv 2012 [31]: of 35 patients with EGFR amplification, 1 of 11 EGFRvIII-positive and 5 of 24 EGFRvIII-negative patients had IDH mutations; Weller, 2014 [43]: all the EGFRvIII-positive patients were IDH-wildtype, but 9% of the EGFRvIII-negative patients had IDH mutations; Sepúlveda-Sánchez 2017 [38]: among 19 EGFRvIII-positive patients, 2 had IDH mutations, while none of the 30 EGFRvIII-negative patients had IDH mutations. Eighteen studies did not specify the IDH status (no testing or unclear inclusion of IDH-mutant cases), while 11 studies explicitly included only IDH-wildtype GBM patients. After excluding the three studies that included confirmed IDH-mutant cases, we reanalyzed the data and found that the results remained unchanged (Appendix A). Despite the use of comprehensive search terms, a substantial number of studies were excluded based on predefined eligibility criteria, which may have introduced a potential selection bias. Although this study included a larger number of articles compared to previous analyses, the significant heterogeneity among the included studies remains a notable concern. Hence, further research incorporating high-quality studies is warranted to obtain more reliable and robust conclusions. Our analysis indicates poorer prognoses among patients in the Americas. Further high-quality studies are warranted to more comprehensively evaluate its prognostic significance in populations from other regions. Our analysis also suggests that the presence of the EGFRvIII mutation in patients with recurrent glioblastoma is associated with a worse prognosis. However, it is crucial to consider the specific status of this mutation, such as whether it has been present since the initial diagnosis or emerged during second-line treatment. Moreover, during treatment, drugs may interact with prognostic biomarkers, potentially affecting the accurate assessment of their prognostic value. Additionally, due to the limited number of studies, we included two studies on EGFR alterations and four studies on EGFR expression (protein-level expression), which do not strictly conform to the definition of EGFR amplification as an increase in the EGFR copy number. Moreover, most of the included studies were retrospective and provided limited patient data, restricting the scope for a more comprehensive analysis. For instance, critical factors, such as dynamic changes in the EGFRvIII mutation status during treatment or patient age stratification, which could have a significant impact on prognostic evaluation, could not be fully explored.

## 5. Conclusions

Our findings indicate that the EGFRvIII mutation holds a significant prognostic value, particularly for patients in the Americas and those with recurrent glioblastoma. We propose that integrating prognostic biomarkers with ethnic and regional factors could improve the precision of prognostic assessments for glioblastoma patients. Furthermore, our analysis suggests that EGFR can serve as an independent prognostic marker for glioblastoma; however, co-expression with the EGFRvIII mutation was not conclusively associated with a poorer prognosis. These findings warrant further validation through high-quality, region-specific studies to strengthen their generalizability and clinical relevance.

## Figures and Tables

**Figure 1 ijms-26-03539-f001:**
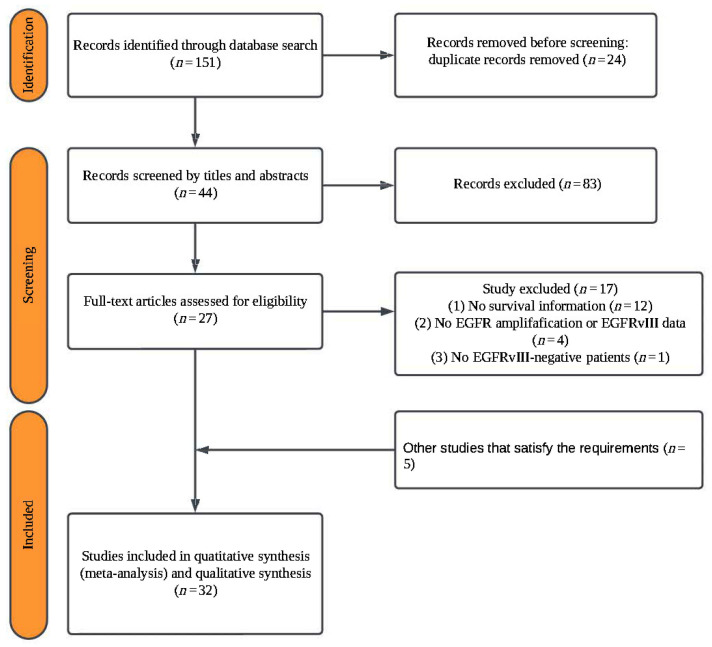
A PRISMA flow chart of the study selection process.

**Figure 2 ijms-26-03539-f002:**
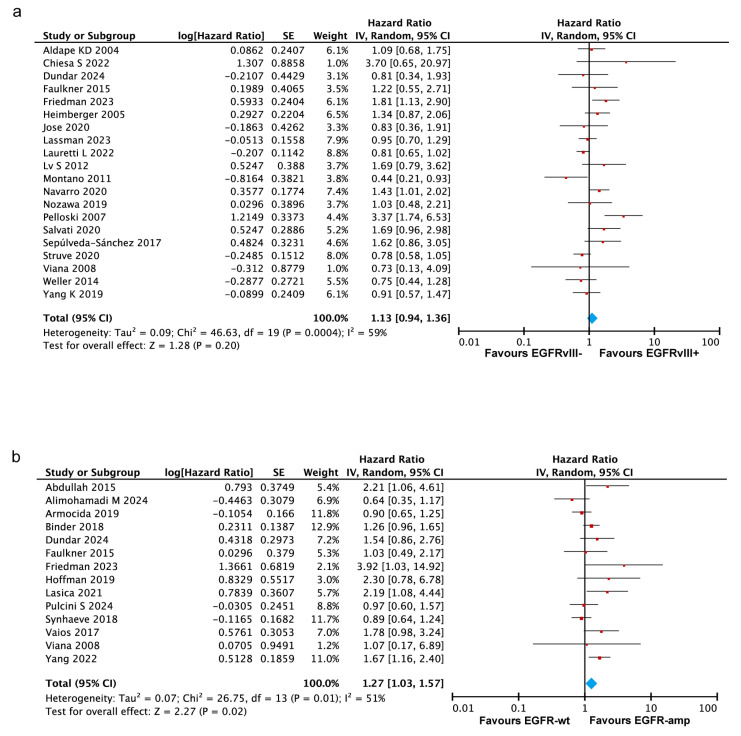
Forest plots illustrating the association between the EGFRvIII mutation or EGFR amplification and the overall survival in GBM patients. Random–effects model was used for analysis. (**a**) Forest plot showing no significant association between EGFRvIII status and OS in GBM patients. (**b**) Forest plot demonstrating that EGFR amplification is associated with poorer OS in GBM patients. In both panels, the size of the red squares represents the relative weight of each study in the overall comparison. The blue diamond at the bottom summarizes the pooled effect estimate, including the HR and the 95% CI across all the included studies. HR values of >1.0 indicate a higher risk of mortality for patients with the EGFRvIII mutation or EGFR amplification compared to those without, while values of <1.0 indicate the opposite. Abbreviations: SE, standard error; CI, confidence interval [11,15,16,17,18,19,20,21,22,23,24,26,27,28,29,31,32,33,34,35,36,37,38,39,40,41,42,43,44].

**Figure 3 ijms-26-03539-f003:**
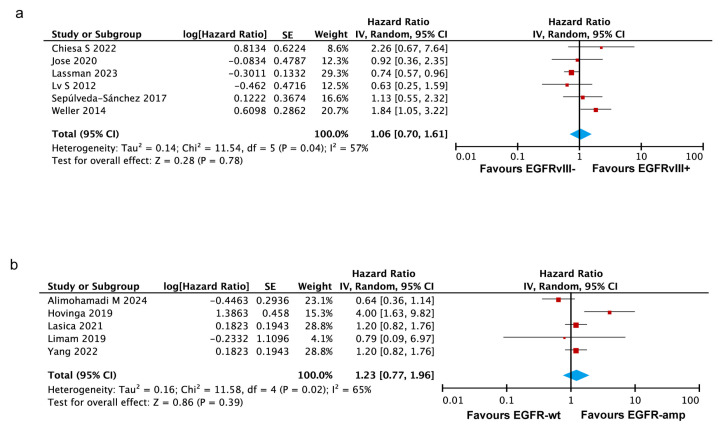
Forest plots illustrating the association of the EGFRvIII mutation and EGFR amplification with progression-free survival in GBM patients. Random-effects model was used for analysis. (**a**) Forest plot showing no significant association between the EGFRvIII mutation and PFS. (**b**) Forest plot showing no significant association between EGFR amplification and PFS. In both panels, the size of the red squares represents the relative weight of each study in the overall comparison. The blue diamond at the bottom summarizes the pooled effect estimate, including the HR and 95% CI across all the included studies. HR values of >1.0 indicate a higher risk of progression for patients with the EGFRvIII mutation or EGFR amplification compared to those without, while values of <1.0 indicate the opposite. Abbreviations: SE, standard error; CI, confidence interval [11,19,25,26,27,28,30,31,38,43,44].

**Figure 4 ijms-26-03539-f004:**
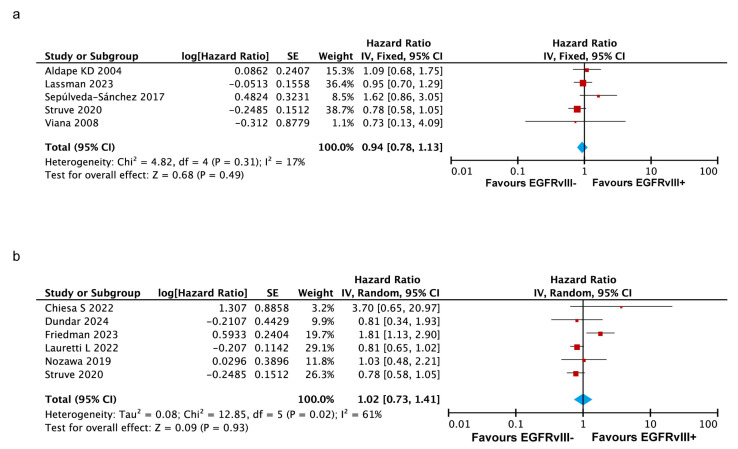
Forest plots illustrating the association of genetic alterations and tumor characteristics with overall survival in GBM patients. (**a**) Forest plot showing no significant association between EGFR amplification co-occurring with EGFRvIII mutation and OS. Random-effects model was used for analysis. (**b**) Forest plot showing no significant association between IDH wild-type and OS in GBM patients. In both panels, the size of the red squares represents the relative weight of each study in the overall comparison. The blue diamond at the bottom summarizes the pooled effect estimate, including the HR and the 95% CI across all the included studies. Fixed-effects model was used for analysis [16,19,20,22,28,29,34,38,39,42].

**Figure 5 ijms-26-03539-f005:**
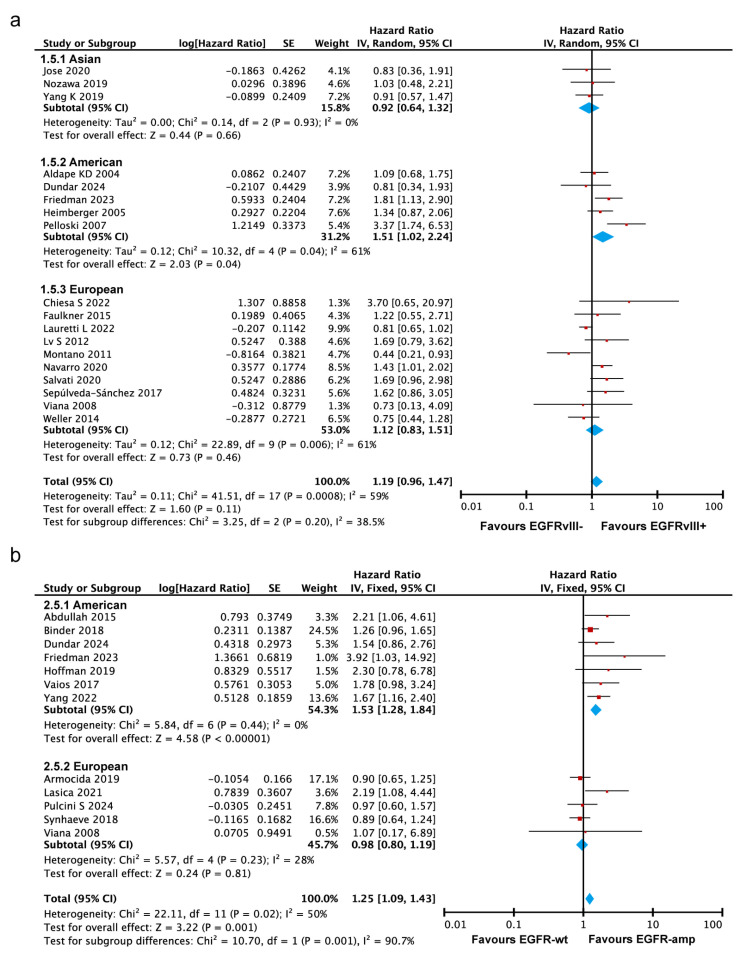
Subgroup analyses based on geographic populations and tumor characteristics. (**a**) Forest plot showing that the EGFRvIII mutation is associated with poorer survival in the Americas but not in Asia or Europe. Random-effects model was used for analysis. (**b**) A forest plot showing that EGFR amplification has a poor prognostic impact in the Americas but not in Asia or Europe. In both panels, the size of the red squares represents the relative weight of each study in the overall comparison. The blue diamond at the bottom summarizes the pooled effect estimate, including the HR and the 95% CI across all the included studies. Fixed-effects model was used for analysis [10,15,16,17,18,19,20,21,22,23,24,26,27,29,31,32,33,34,35,36,37,38,40,41,42,43,44].

**Figure 6 ijms-26-03539-f006:**
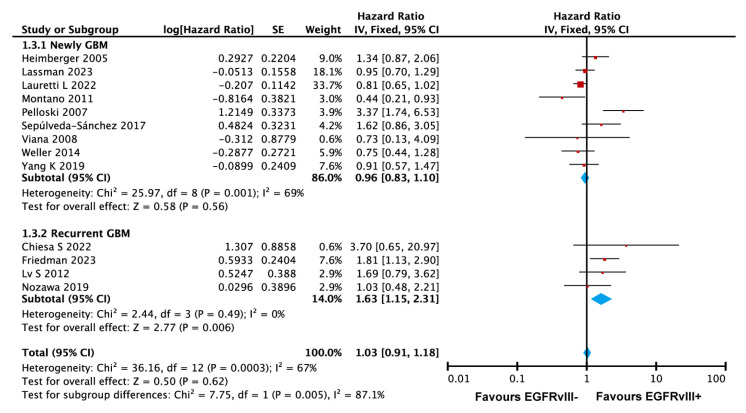
Forest plot showing that EGFRvIII mutation is associated with poorer survival in recurrent GBM. In both panels, the size of the red squares represents the relative weight of each study in the overall comparison. The blue diamond at the bottom summarizes the pooled effect estimate, including the HR and the 95% CI across all the included studies. Fixed-effects model was used for analysis [19,22,23,28,29,31,32,34,35,38,42,43,44].

**Table 1 ijms-26-03539-t001:** The characteristics of the studies included in the analysis of the association between the EGFRvIII mutation and the patient prognosis.

							Patients Number		Outcomes (HR 95% CI)	
Study Name	EGFR-amp *	Year	Country	Recurrent/Newly Diagnosed GBM	Intervention	Study Design	EGFRvIII+	EGFRvIII–	OS	PFS
Aldape 2004 [16]	Yes	NA	U.S.A.	NA	NA	Retrospective Cohort Study	22	77	1.09 [0.68–1.76]	NA
Chiesa 2022 [19]	No	NA	Italy	Recurrent GBM	Regorafenib	Prospective Cohort Study	14	14	3.695 [0.651–20.963]	2.2556 [0.666–9.814]
Dundar 2024 [20]	No	2020–2021	U.S.A.	NA	NA	Retrospective Cohort Study	19	25	0.81 [0.34–1.96]	NA
Faulkner 2015 [21]	No	NA	U.K.	NA	NA	Retrospective Cohort Study	16	35	1.22 [0.55–2.69]	NA
Friedman 2023 [22]	No	2014–2019	U.S.A.	Recurrent GBM	Bev/Niv/Pem	Retrospective Cohort Study	NA	NA	1.81 [1.13–2.89]	NA
Heimberger 2005 [23]	No	NA	U.S.A.	New GBM	RT	Prospective Cohort Study	61	44	1.34 [0.87–2.07]	NA
Jose 2020 [26]	NA	2014–2015	India	NA	RT + TMZ	Retrospective Cohort Study	23	17	0.83 [0.36–1.94]	0.92 [0.36–2.30]
Lassman 2023 [28]	Yes	2015–2018	Global	New GBM	RT + TMZ	Phase III Randomized Clinical Trial	168	148	0.95 [0.70–1.29]	0.74 [0.57–0.97]
Lauretti 2022 [29]	NA	2012–2017	Italy	New GBM	RT + TMZ	Prospective Cohort Study	184	171	0.813 [0.650–1.016]	NA
Lv 2012 [31]	No	NA	Belgium	Recurrent GBM	Cetuximab	Retrospective Cohort Study	11	24	1.69 [0.79–3.61]	0.63 [0.25–1.60]
Montano 2011 [32]	NA	NA	Italy	New GBM	RT + TMZ	Prospective Cohort Study	32	41	0.442 [0.209–0.714]	NA
Navarro 2020 [33]	NA	1995–2010	Spain	NA	RT + TMZ	Retrospective Cohort Study	45	83	1.43 [1.01–2.03]	NA
Nozawa 2019 [34]	NA	NA	Japan	Recurrent GBM	RT/TMZ/Bev	Cohort Study	13	54	1.03 [0.48–2.21]	NA
Pelloski 2007 [35]	NA	1992–2003	U.S.A.	New GBM	RT	Cohort Study	84	184	3.37 [1.74,6.50]	NA
Salvati 2020 [37]	NA	2013–2017	Italy	NA	RT + TMZ	Retrospective Cohort Study	61	61	1.69 [0.96–2.98]	NA
Sepúlveda-Sánchez 2017 [38]	Yes	2012–2015	Spain	New GBM	Dacomitinib	Open-Label Phase II Clinical Trial	19	30	1.62 [0.86–3.07]	1.13 [0.55–2.31]
Struve 2020 [39]	Yes	NA	NA	NA	RT + TMZ	Prospective Cohort Study	92	244	0.78 [0.58–1.06]	NA
Viana-Pereira 2008 [42]	Yes	NA	Portugal	New GBM	NA	Prospective Cohort Study	8	23	0.732 [0.131–4.095]	NA
Weller 2014 [43]	No	NA	Germany	New GBM	RT + TMZ	Prospective Cohort Study	85	99	0.75 [0.44–1.29]	1.84 [1.05–3.21]
Yang 2019 [44]	NA	2011–2015	China	New GBM	RT + TMZ	Retrospective Cohort Study	73	167	0.914 [0.570–1.467]	NA

The studies are labeled by the last name of the first author and presented in alphabetical order, along with the publication year. Abbreviations: RT = radiation therapy; TMZ = temozolomide; Bev/Niv/Pem = bevacizumab/nivolumab/pembrolizumab; NA = not available; EGFRvIII+/− = EGFR variant III positive/negative; EGFR-amp = EGFR amplification; EGFR-wt = EGFR wildtype; OS = overall survival; PFS = progression-free survival; HR = hazard ratio; 95% CI = 95% confidence interval. *: “Yes” indicates that all the patients’ tumors simultaneously exhibit EGFR amplification and the EGFRvIII mutation, while “No” indicates that the patients included in the analysis do not exhibit both alterations simultaneously.

**Table 2 ijms-26-03539-t002:** The characteristics of the studies included in the analysis of the association between EGFR amplification and patient prognosis.

						Patients Number		Outcomes (HR 95% CI)	
Study Name	Year	Country	Recurrent/Newly Diagnosed GBM	Intervention	Study Design	EGFR-amp	EGFR-wt	OS	PFS
Abdullah 2015 [15]	2003–2013	U.S.A.	Recurrent GBM	Surgical resection	Retrospective Cohort Study	20	27	2.21 [1.06–4.60]	NA
Alimohamadi 2024 [11]	2019–2021	Iran	New GBM	RT/TMZ	Prospective Cohort study	17	14	0.64 [0.35–1.16]	0.64 [0.36–1.14]
Armocida 2019 [17]	2014–2016	Italy	New GBM	RT/TMZ	Retrospective Cohort Study	9	21	0.90 [0.65–1.24]	NA
Binder 2018 [18]	2013–2016	U.S.A.	New GBM	RT/TMZ	Retrospective Cohort Study	97	160	1.26 [0.96–1.66]	NA
Dundar 2024 [20]	2020–2021	U.S.A.	NA	NA	Retrospective Cohort Study	127	233	1.54 [0.86–2.75]	NA
Faulkner 2015 [21]	NA	U.K.	NA	NA	Retrospective Cohort Study	22	29	1.03 [0.49–2.18]	NA
Friedman 2023 [22]	2014–2019	U.S.A.	Recurrent GBM	Bev/Niv/Pem	Retrospective Cohort Study	5	10	3.92 [1.03–14.9]	NA
Hoffman 2019 [24]	2014–2018	U.S.A.	New GBM	RT/TMZ	Retrospective Cohort Study	10	18	2.30 [0.78–6.73]	NA
Hovinga 2019 [25]	2006–2014	U.S.A.	Recurrent GBM	Bev	Retrospective Cohort Study	28	37	NA	4.00 [1.63–9.77]
Lasica 2021 [27]	2010–2019	U.K.	NA	NA	Retrospective Cohort Study	47	149	2.19 [1.08–4.44]	1.20 [0.82–1.73]
Limam 2019 [30]	2009–2015	Tunisia	NA	NA	Retrospective Cohort Study	59	15	NA	0.792 [0.090–6.955]
Pulcini 2024 [36]	2013–2022	France	NA	RT/TMZ	Retrospective Cohort Study	30	43	0.97 [0.6–1.57]	NA
Synhaeve 2018 [40]	2013–2017	Netherlands	NA	NA	Retrospective Cohort Study	83	97	0.89 [0.64–1.22]	NA
Vaios 2017 [41]	2007–2014	U.S.A.	New GBM	RT/TMZ	Retrospective Cohort Study	37	37	1.779 [0.978–3.247]	NA
Viana-Pereira 2008 [42]	NA	Portugal	New GBM	NA	Retrospective Cohort Study	10	9	1.073 [0.167–6.885]	NA
Yang 2022 [10]	2015–2020	U.S.A.	New GBM	RT/TMZ	Retrospective Cohort Study	66	185	1.67 [1.16–2.42]	1.20 [0.82–1.73]

The studies are labeled by the last name of the first author and presented in alphabetical order, along with the publication year. Abbreviations: RT = radiation therapy; TMZ = temozolomide; Bev/Niv/Pem = bevacizumab/nivolumab/pembrolizumab; NA = not available; EGFRvIII+/− = EGFR variant III positive/negative; EGFR-amp = EGFR amplification; EGFR-wt = EGFR wildtype; OS = overall survival; PFS = progression-free survival; HR = hazard ratio; 95% CI = 95% confidence interval.

**Table 3 ijms-26-03539-t003:** A quality assessment of the studies used in this meta-analysis, based on the Newcastle–Ottawa scale.

	Selection					Outcome Assessment			
Study	1	2	3	4	Comparability	1	2	3	Score
Abdullah 2015 [15]	★	★	★	☆	★★	★	★	☆	7
Aldape 2004 [16]	★	★	☆	☆	★★	★	★	★	7
Alimohamadi 2024 [11]	★	★	☆	★	★★	★	★	☆	7
Armocida 2019 [17]	★	★	☆	☆	★★	☆	★	☆	5
Binder 2018 [18]	★	★	★	☆	★	★	★	☆	6
Chiesa 2022 [19]	★	★	☆	★	★★	★	★	★	8
Dundar 2024 [20]	★	★	☆	☆	★★	★	★	☆	6
Faulkner 2015 [21]	★	★	☆	☆	★★	★	★	☆	6
Friedman 2023 [22]	☆	★	☆	☆	★★	★	★	★	6
Heimberger 2005 [23]	★	★	★	☆	★★	★	★	☆	7
Hoffman 2019 [24]	★	★	★	☆	★★	★	★	☆	7
Hovinga 2019 [25]	★	★	★	☆	★★	★	★	☆	7
Jose 2020 [26]	★	★	☆	☆	★★	★	★	★	7
Lasica 2021 [27]	☆	★	★	☆	★★	★	★	☆	6
Lauretti 2022 [29]	★	★	☆	★	★★	☆	★	☆	6
Limam 2019 [30]	★	★	★	☆	★	★	★	☆	6
Lv 2012 [31]	★	★	☆	☆	★★	★	★	★	7
Montano 2011 [32]	★	★	☆	★	★★	☆	★	★	7
Navarro 2020 [33]	★	★	☆	☆	★	☆	★	★	5
Nozawa 2019 [34]	★	★	☆	☆	★★	☆	★	★	6
Pelloski 2007 [35]	★	★	☆	☆	★★	☆	★	☆	5
Pulcini 2024 [36]	★	★	☆	☆	★★	★	★	☆	6
Salvati 2020 [37]	★	★	☆	☆	★★	★	★	☆	6
Struve 2020 [39]	★	★	☆	☆	★	★	★	☆	5
Synhaeve 2018 [40]	★	★	★	★	★	★	★	★	8
Vaios 2017 [41]	★	★	★	☆	★★	★	★	☆	7
Viana-Pereira 2008 [42]	★	★	★	☆	★★	★	★	★	8
Weller 2014 [43]	★	★	☆	★	★★	☆	★	☆	6
Yang 2022 [10]	★	★	★	☆	★★	★	★	★	8
Yang 2019 [44]	☆	★	☆	☆	★★	★	★	☆	5

★ = one point awarded; ☆ = no point awarded.

**Table 4 ijms-26-03539-t004:** A quality assessment of the studies used in this meta-analysis, based on the Cochrane Collaboration Risk of Bias Tool 2.

Study Name	Randomization Process	Bias Due to Deviations from Intended Interventions	Missing Outcome Data	The Bias in the Measurement of the Outcome	The Bias in the Selection of the Reported Result
Lassman 2023 [28]	Low Risk	Low Risk	Low Risk	Low Risk	Low Risk
Sepúlveda-Sánchez 2017 [38]	High Risk	High Risk	Moderate risk	Moderate risk	Moderate risk

## Data Availability

The data that support the findings of this study are available on request from the corresponding author (Y.F.).

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
