# Peer review of "The Prognostic Significance of Epidermal Growth Factor Receptor Amplification and Epidermal Growth Factor Receptor Variant III Mutation in Glioblastoma: A Systematic Review and Meta-Analysis with Implications for Targeted Therapy"

_ijms, 2025, doi:10.3390/ijms26083539_

Round 1

Reviewer 1 Report

Comments and Suggestions for Authors

In this review article, the authors summarized recent reports on the clinical studies of EGFR amplification and EGFRvIII mutation in glioblastoma patients. The EGFR amplification is significantly correlated with worse OS, whereas the EGFRvIII mutation may be associated with poorer outcomes, particularly in recurrent GBM patients. The review is mostly well written and organized. The reviewer has only a few comments.

Comments.

  1. In the case of EGFRvIII mutation, the copy number of this mutant gene is gradually increased in the recurrent GBM patients?
  2. In the case of EGFR amplification, the American GBM patients show poor prognosis, but not for the European patients. What kind of reasons are considered?

Author Response

Comments 1: “In the case of EGFRvIII mutation, the copy number of this mutant gene is gradually increased in the recurrent GBM patients?”

Our response: We greatly appreciate the reviewer’s thoughtful question. Unfortunately, our study primarily focused on the presence or absence of EGFRvIII mutation within tumors rather than the copy number of the mutant gene, and most of the included studies did not provide detailed copy number data.

However, some studies have demonstrated that EGFR alterations, including EGFRvIII mutations, can either persist or loss in recurrent GBM1. For instance, Martin J. van den Bent et al. utilized RT-qPCR to evaluate EGFR and EGFRvIII expression in 89 recurrent GBM patients2. They found that EGFR status remained stable in 84% of the tumors, while EGFRvIII expression persisted in 79%. However, approximately half of the tumors expressing EGFRvIII at initial diagnosis lost this expression upon recurrence2. Furthermore, O'Rourke DM et al. treated 10 recurrent glioblastoma patients with intravenous EGFRvIII-CAR T cells3. Among the seven evaluated patients, five lost EGFRvIII expression. This finding indicates the need to consider tumor microenvironment changes and antigen heterogeneity in recurrent GBM treatment.

Our analysis suggests that EGFRvIII mutation in recurrent GBM is linked to poorer prognosis, supporting the association with increased tumor malignancy and treatment resistance4,5. However, the relationship between patient prognosis and the temporal persistence or copy number variation of EGFRvIII in recurrent tumors did not specifically address in this analysis. Recognizing its potential significance, future studies should consider the dynamic shifts in molecular subtypes and the simultaneous activation of multiple oncogenic pathways to better understand this correlation6,7.

Comments 2: “In the case of EGFR amplification, the American GBM patients show poor prognosis, but not for the European patients. What kind of reasons are considered?”

Our response: Thank you for your insightful question. Global epidemiological data indicate that GBM incidence is highest in North America and Europe, with non-Hispanic whites in the United States exhibiting the highest prevalence8-10, suggesting potential racial differences in GBM susceptibility.

  These differences may derive from genetic factors, molecular profiles, treatment strategies, and environmental influences. However, due to the limited number of studies and insufficient patient information, we were unable to conduct an in-depth analysis of these risk factors. Future research will focus on region-specific meta-analyses with detailed data on patient demographics, treatment protocols, and tumor molecular characteristics. Integrating multi-omics approaches, such as transcriptomics and proteomics, may further elucidate the biological mechanisms underlying these regional differences.

References:

1          Cioca, A., Olteanu, E. G., Gisca, M. D., Morosanu, C. O., Marin, I. & Florian, I. S. Expression of EGFR in Paired New and Recurrent Glioblastomas. Asian Pac J Cancer Prev 17, 4205-4208 (2016).

2          van den Bent, M. J. et al. Changes in the EGFR amplification and EGFRvIII expression between paired primary and recurrent glioblastomas. Neuro Oncol 17, 935-941 (2015). https://doi.org:10.1093/neuonc/nov013

3          O'Rourke, D. M. et al. A single dose of peripherally infused EGFRvIII-directed CAR T cells mediates antigen loss and induces adaptive resistance in patients with recurrent glioblastoma. Sci Transl Med 9 (2017). https://doi.org:10.1126/scitranslmed.aaa0984

4          Hovinga, K. E. et al. EGFR amplification and classical subtype are associated with a poor response to bevacizumab in recurrent glioblastoma. J Neurooncol 142, 337-345 (2019). https://doi.org:10.1007/s11060-019-03102-5

5          Tong, F. et al. MUC1 promotes glioblastoma progression and TMZ resistance by stabilizing EGFRvIII. Pharmacol Res 187, 106606 (2023). https://doi.org:10.1016/j.phrs.2022.106606

6          Neftel, C. et al. An Integrative Model of Cellular States, Plasticity, and Genetics for Glioblastoma. Cell 178, 835-849 e821 (2019). https://doi.org:10.1016/j.cell.2019.06.024

7          Park, A. K., Kim, P., Ballester, L. Y., Esquenazi, Y. & Zhao, Z. Subtype-specific signaling pathways and genomic aberrations associated with prognosis of glioblastoma. Neuro Oncol 21, 59-70 (2019). https://doi.org:10.1093/neuonc/noy120

8          Tan, A. C., Ashley, D. M., Lopez, G. Y., Malinzak, M., Friedman, H. S. & Khasraw, M. Management of glioblastoma: State of the art and future directions. CA Cancer J Clin 70, 299-312 (2020). https://doi.org:10.3322/caac.21613

9          Dubrow, R. & Darefsky, A. S. Demographic variation in incidence of adult glioma by subtype, United States, 1992-2007. BMC Cancer 11, 325 (2011). https://doi.org:10.1186/1471-2407-11-325

10        Cancer, W. H. O.-I. A. f. R. o. Cancer Today, <https://gco.iarc.who.int/today/en> (2022).

Reviewer 2 Report

Comments and Suggestions for Authors

Dear Autors,

Your review on the prognostic significance of EGFR amplification and EGFRvII mutation in glioblastoma is a well designed and comprehensive study not available so for. You did adhere well to predefined criteria to come up with a reliable meta-analysis. Nevertheless, there are some points that you may consider.

You included glioblastoma (GBM) studies from 2004 to 2024. This has to be done with great care, as the WHO definition of GBM has changed within this time significantly. At least 2-times with a very strong impact on genetic background and clinical outcome. This is also reflected in your manuscript as you mention the IDH mutation to be a biomarker in GBM. By definition of WHO 2021 any tumor with IDH mutation is no GBM anymore. So, excluding all IDH mutant tumors from your analysis would be mandatory in this respect. Also, the subclassing of GBMs is totally ignored in your review, which is understandable as this information is not available for most studies. But those subclasses are shown to have impact on clinical behaviour and by this influence your outcome data. Although this cannot be included in your analysis, it should at least be discussed.  

Line 11: “most aggressive” is not appropriate here. It is true for brain tumos, but not in general.

Line 48: “whenever probable” does not have the meaning which is intended here. It should be “whenever possible”

Line 55: EGFR is no oncogene by definition. As long as it is not amplified or mutated it is a proto-oncogene.

Line 157: “was” should be “were”

Line 409: I am not aware of any “more aggressive treatment strategies” in GBM than the ones you mentioned already.

Line 452-453: you refer to “Our published study…”, but refer to 66, which included none of the authors of this manuscript. Please check the correctness of this reference.

Best regards

Comments on the Quality of English Language

There are several smaller errors (line 11, 48, 55, 157,...) that have to be corrected, but nothing major.

Author Response

Q1: “You included glioblastoma (GBM) studies from 2004 to 2024. This has to be done with great care, as the WHO definition of GBM has changed within this time significantly. At least 2-times with a very strong impact on genetic background and clinical outcome. This is also reflected in your manuscript as you mention the IDH mutation to be a biomarker in GBM. By definition of WHO 2021 any tumor with IDH mutation is no GBM anymore. So, excluding all IDH mutant tumors from your analysis would be mandatory in this respect.”

Our response: We sincerely thank the reviewer for the valuable comments and constructive feedback, which have greatly helped us improve the quality of our manuscript. Our included studies span from 2004 to 2024, during which the WHO definition of GBM underwent two significant changes. The first was in the 2016 WHO classification, where GBM was divided into IDH-wildtype and IDH-mutant subtypes. The second was in the 2021 WHO classification, where IDH-mutant tumors were reclassified as anaplastic astrocytoma (WHO grade IV) instead of GBM.

Among our included studies, three reported a small proportion of IDH-mutant cases (<10%) in GBM patients with EGFR amplification or EGFRvIII mutation: Lv S, 2012: Of 35 patients with EGFR amplification, 1 of 11 EGFRvIII-positive and 5 of 24 EGFRvIII-negative patients had IDH mutations; Weller, 2014: All EGFRvIII-positive patients were IDH-wildtype, but 9% of EGFRvIII-negative patients had IDH mutations; Sepúlveda S, 2017: Among 19 EGFRvIII-positive patients, 2 had IDH mutations, while none of the 30 EGFRvIII-negative patients had IDH mutations. Eighteen studies did not specify IDH status (no testing or unclear inclusion of IDH-mutant cases), while 11 studies explicitly included only IDH-wildtype GBM patients. After excluding the three studies that included confirmed IDH-mutant cases, we reanalyzed the data and found that the results remained unchanged. The updated pooled analysis showed that EGFRvIII mutation had no significant impact on OS (pooled HR = 1.12, 95% CI: 0.91–1.37, p = 0.28), which has been included in Figure S5a. This result is consistent with our previous analysis (pooled HR = 1.13, 95% CI: 0.94–1.36, p = 0.20). Similarly, there was no significant effect on PFS (pooled HR = 0.91, 95% CI: 0.54–1.54, p = 0.73) (Figure S5b.), consistent with the initial analysis (pooled HR = 1.06, 95% CI: 0.70–1.61, p = 0.78). Our subgroup analysis results also remained consistent with the results of our initial analysis in this study. The impact of EGFRvIII mutation on OS across people with different geographic regions remained unchanged (pooled HR = 1.19, 95% CI: 0.93–1.51, p = 0.16), including the observation of poorer prognosis in patients from the Americas (Figure S5c.). Additionally, EGFRvIII mutation showed no significant impact on OS in newly diagnosed vs. recurrent GBM patients (pooled HR=1.02, 95% CI: 0.88–1.17, p=0.01) (Figure S5d.), consistent with the initial findings. Notably, the analysis of EGFR alteration in relation to patients’ prognosis was not affected, as the three excluded studies were not part of that analysis.

Although our updated analysis aligns with previous conclusions, it is important to note that 18 studies lacked clear IDH mutation status in both analyses. This could have led to the inadvertent inclusion of IDH-mutant cases, potentially overestimating the prognostic impact. In future studies or meta-analyses on GBM prognostic markers, we will strict exclusion of IDH-mutant tumors to ensure more reliable conclusions.

Q2: “Also, the subclassing of GBMs is totally ignored in your review, which is understandable as this information is not available for most studies. But those subclasses are shown to have impact on clinical behaviour and by this influence your outcome data. Although this cannot be included in your analysis, it should at least be discussed.”

Our response: We have reviewed the relevant literature and added content to the discussion section regarding the impact of GBM classification on patient prognosis. We have also revised the relevant data analysis sections. All modifications have been marked in red in the revised manuscript: “In the latest WHO classification of CNS tumors, glioblastoma is defined as an IDH-wildtype grade IV diffuse glioma, which can be further subdivided into molecular subtypes based on specific characteristics[1]. According to extensive transcriptome profiling and high-throughput next-generation sequencing, GBM is commonly classified into four subtypes with distinct molecular features, clinical behaviors, and prognoses: classical (CL), mesenchymal (MES), proneural (PN), and neural (which is often excluded due to the belief that it represents normal neuronal lineage contamination). MES tumors are characterized by high NF1 expression, while PN tumors frequently exhibit PDGFRA mutations and contain tumor cells similar to oligodendrocyte progenitors and neural progenitors. CL tumors mainly consist of astrocyte-like cells and are frequently associated with EGFR and EGFRvIII mutations[2, 3]. CL-type GBM patients tend to respond relatively well to chemoradiotherapy but generally have poor prognoses[4]. EGFR alterations play a pivotal role in driving the differentiation of GBM tumors toward the CL subtype. As early as 1998, Sibilia M et al. demonstrated that EGFR could promote astrocytes proliferation and differentiation in the brain[5]. Recently, Wei et al. further confirmed that BCAR4 enhances glioma progression by activating the EGFR/PI3K/AKT pathway, promoting astrocytes transformation and tumor proliferation[6]. In a retrospective study of 80 CL-type recurrent GBM patients, Hovinga KE et al. found that EGFR mutations were associated with poorer outcomes following bevacizumab treatment[7]. However, another study by Park AK et al. Involving 23 recurrent GBM patients reported no significant correlation between EGFR mutations and OS (HR = 0.84, p = 0.585)[8]. Our analysis indicates that EGFRvIII mutation is associated with poorer prognosis in recurrent glioblastoma. Based on the association between EGFR alterations and CL subtype, our findings are most likely relevant to the prognostic evaluation of EGFRvIII mutation in CL subtype recurrent GBM patients. However, its prognostic significance for other GBM subtypes remains uncertain. Among the studies we included, only one speculated that these patients might belong to the MES subtype, based on MAPK pathway activation in NGS results[9]. The other studies lacked detailed GBM subtypes information. Further analysis of the association between highly mutated oncogenes and the prognosis of different GBM subtypes remains necessary. Future studies should consider the potential intratumoral heterogeneity of molecular subtypes, investigate subtype specific oncogenic alteration, and identify substitute markers. These factors are crucial for accurately predicting prognosis and optimizing treatment strategies for GBM patients with distinct molecular subtypes.”

Q3: “Line 11: “most aggressive” is not appropriate here. It is true for brain tumors, but not in general.”

Our response: Thank you for pointing out this issue. This sentence has been revised to “Glioblastoma (GBM) is the most aggressive and heterogeneous neoplasm among central nervous system tumors, with a dismal prognosis and a high recurrence rate.”

Q4: Line 48: “whenever probable” does not have the meaning which is intended here. It should be “whenever possible”

Our response: Thank you for pointing out this issue. We have revised “whenever probable” to “whenever possible”.

Q5: Line 55: EGFR is no oncogene by definition. As long as it is not amplified or mutated it is a proto-oncogene.

Our response: Thanks for pointing out this issue. We have changed the “potent oncogene” to “proto-oncogene.”

Q6: Line 157: “was” should be “were”

Our response: Thank you for pointing out this mistake. We have revised line 157 and corrected “was” to “were.”

Q7: Line 409: I am not aware of any “more aggressive treatment strategies” in GBM than the ones you mentioned already.

Our response: Thank you for pointing out this issue. The sentences have been revised to This finding highlights the potential of EGFRvIII as a diagnostic tool for identifying high-risk recurrent GBM cases, which may benefit from more personalized and molecularly targeted therapies, such as EGFRvIII-directed CAR-T therapy.

Q8: Line 452-453: you refer to “Our published study…”, but refer to 66, which included none of the authors of this manuscript. Please check the correctness of this reference.

Our response: Thank you for pointing out this mistake. We accidentally cited the wrong reference. The sentence “Our published study also demonstrated that Osimertinib effectively inhibits EGFR expression and the activation of associated pathways [66]” has now been corrected.

References: 

[1] Louis, D. N.; Perry, A.; Wesseling, P.; Brat, D. J.; Cree, I. A.; Figarella-Branger, D.; Hawkins, C.; Ng, H. K.; Pfister, S. M.; Reifenberger, G.; et al. The 2021 WHO Classification of Tumors of the Central Nervous System: a summary. Neuro Oncol 2021, 23 (8), 1231-1251. DOI: 10.1093/neuonc/noab106  From NLM Medline.

[2] Verhaak, R. G.; Hoadley, K. A.; Purdom, E.; Wang, V.; Qi, Y.; Wilkerson, M. D.; Miller, C. R.; Ding, L.; Golub, T.; Mesirov, J. P.; et al. Integrated genomic analysis identifies clinically relevant subtypes of glioblastoma characterized by abnormalities in PDGFRA, IDH1, EGFR, and NF1. Cancer Cell 2010, 17 (1), 98-110. DOI: 10.1016/j.ccr.2009.12.020  From NLM Medline.

[3] Stichel, D.; Ebrahimi, A.; Reuss, D.; Schrimpf, D.; Ono, T.; Shirahata, M.; Reifenberger, G.; Weller, M.; Hanggi, D.; Wick, W.; et al. Distribution of EGFR amplification, combined chromosome 7 gain and chromosome 10 loss, and TERT promoter mutation in brain tumors and their potential for the reclassification of IDHwt astrocytoma to glioblastoma. Acta Neuropathol 2018, 136 (5), 793-803. DOI: 10.1007/s00401-018-1905-0  From NLM Medline.

[4] Li, H.; He, J.; Li, M.; Li, K.; Pu, X.; Guo, Y. Immune landscape-based machine-learning-assisted subclassification, prognosis, and immunotherapy prediction for glioblastoma. Front Immunol 2022, 13, 1027631. DOI: 10.3389/fimmu.2022.1027631  From NLM Medline.

[5] Sibilia, M.; Steinbach, J. P.; Stingl, L.; Aguzzi, A.; Wagner, E. F. A strain-independent postnatal neurodegeneration in mice lacking the EGF receptor. Embo j 1998, 17 (3), 719-731. DOI: 10.1093/emboj/17.3.719  From NLM.

[6] Wei, L.; Yi, Z.; Guo, K.; Long, X. Long noncoding RNA BCAR4 promotes glioma cell proliferation via EGFR/PI3K/AKT signaling pathway. J Cell Physiol 2019, 234 (12), 23608-23617. DOI: 10.1002/jcp.28929  From NLM.

[7] Hovinga, K. E.; McCrea, H. J.; Brennan, C.; Huse, J.; Zheng, J.; Esquenazi, Y.; Panageas, K. S.; Tabar, V. EGFR amplification and classical subtype are associated with a poor response to bevacizumab in recurrent glioblastoma. J Neurooncol 2019, 142 (2), 337-345. DOI: 10.1007/s11060-019-03102-5  From NLM Medline.

[8] Park, A. K.; Kim, P.; Ballester, L. Y.; Esquenazi, Y.; Zhao, Z. Subtype-specific signaling pathways and genomic aberrations associated with prognosis of glioblastoma. Neuro Oncol 2019, 21 (1), 59-70. DOI: 10.1093/neuonc/noy120  From NLM Medline.

[9] Chiesa, S.; Mangraviti, A.; Martini, M.; Cenci, T.; Mazzarella, C.; Gaudino, S.; Bracci, S.; Martino, A.; Della Pepa, G. M.; Offi, M.; et al. Clinical and NGS predictors of response to regorafenib in recurrent glioblastoma. Sci Rep 2022, 12 (1), 16265. DOI: 10.1038/s41598-022-20417-y (acccessed 2024/11/13/09:20:24). From 4.6.

Round 2

Reviewer 2 Report

Comments and Suggestions for Authors

Thank you for addressing all remarks adequately.